# Automatic real-time analysis and interpretation of arterial blood gas sample for Point-of-care testing: Clinical validation

**Sancho Rodríguez-Villar**[1]*, **Paloma Poza-Hernández**[2], **Sascha Freigang**[3], **Idoia Zubizarreta-Ormazabal**[4], **Daniel Paz-Martín**[2‡], **Etienne Holl**[3‡], **Osvaldo Ceferino Pérez-Pardo**[4‡], **María Sherezade Tovar-Doncel**[2‡], **Sonja Maria Wissa**[3‡], **Bonifacio Cimadevilla-Calvo**[4‡], **Guillermo Tejón-Pérez**[4‡], **Ismael Moreno-Fernández**[5‡], **Alejandro Escario-Méndez**[5‡], **Juan Arévalo-Serrano**[6‡], **Antonio Valentín**[7‡], **Bruno Manuel Do-Vale**[8‡], **Helen Marie Fletcher**[9‡], **Jesús Medardo Lorenzo- Fernández**[10‡]

1 Critical Care Department, King´s College Hospital, London, United Kingdom, 2 Hospital Virgen de la Salud, Servicio de Anestesia y Reanimación, Surgical Intensive Care Unit, Complejo Hospitalario de Toledo, Toledo, Spain, 3 Department of Neurosurgery, Neurosurgical Intensive Care Unit, Medical University of Graz, Graz, Austria, 4 Marqués de Valdecilla University Hospital (HUMV), Servicio de Anestesia y Reanimación, Santander, Cantabria, Spain, 5 Medical Software Specialist at Madrija Company, Toledo, Spain, 6 Internal Medicine Department, Príncipe de Asturias Hospital, Alcalá de Henares, Madrid, Spain, 7 Department of Basic and Clinical Neuroscience, Institute of Psychiatry, Psychology & Neuroscience (IoPPN) Academic Neuroscience Centre, London, United Kingdom, 8 Critical Care Department, Centro Hospitalar Universitário do Porto (CHUP), Porto, Portugal, 9 Executive Nursing Department, King´s College Hospital, London, United Kingdom, 10 External Consultant, Ingeniero de Caminos, Canales y Puertos e Ingeniero Técnico de Obras Públicas

⊙ These authors contributed equally to this work.
‡ These authors also contributed equally to this work.
* sancho.villar@nhs.net

**Data Availability Statement:** In accordance with the Declaration of Helsinki regarding the confidentiality of the patient's information, The

## Abstract

### Background

Point-of-care arterial blood gas (ABG) is a blood measurement test and a useful diagnostic tool that assists with treatment and therefore improves clinical outcomes. However, numerically reported test results make rapid interpretation difficult or open to interpretation. The arterial blood gas algorithm (ABG-a) is a new digital diagnostics solution that can provide clinicians with real-time interpretation of preliminary data on safety features, oxygenation, acid-base disturbances and renal profile. The main aim of this study was to clinically validate the algorithm against senior experienced clinicians, for acid-base interpretation, in a clinical context.

### Methods

We conducted a prospective international multicentre observational cross-sectional study. 346 sample sets and 64 inpatients eligible for ABG met strict sampling criteria. Agreement was evaluated using Cohen's kappa index, diagnostic accuracy was evaluated with sensitivity, specificity, efficiency or global accuracy and positive predictive values (PPV) and negative predictive values (NPV) for the prevalence in the study population.

General Data Protection Regulation (25 May 2018), local applicable regulatory requirements and the ethical approvals for the study, the deidentified study data can be made available to researchers upon request and after approval by the local Ethics Committee (Comité de Ética de la Investigación con Medicamentos (CEIm) de Toledo, Spain; Comité de Ética de la Investigación con Medicamentos (CEIm) de Cantabria, Spain and Vice-Rector for Research and International Affairs, Mag. a Caroline Schober-Trummler on behalf of the Med Uni Graz.) for researchers who meet the criteria for access to confidential data by contacting Comité de Ética de la Investigación con Medicamentos (CEIm) de Toledo, Spain (ref: 05/12/19. Number 461): docenciamir@sescam.jccm.es.

**Funding:** Financial support, including any institutional departmental funds, was not sought for the study. The authors received no specific funding for this work Medical Software Madrija Company provided financial support in the form of a salary to Moreno-Fernández I and Escario-Méndez A, but did not have any additional role in the study design, data collection and analysis, decision to publish, or preparation of the manuscript. The specific roles of these authors are articulated in the 'author contributions' section.

**Competing interests:** The validation study is a collaboration among healthcare professionals: Rodríguez-Villar S, Do-Vale BM and Fletcher HM who hold currently the ABG-a patent and original design; the medical team who run the study in the different centres in Europe: Poza-Hernández P, Freigang S, Zubizarreta-Ormazabal I, Paz-Martín D, &Holl E, Pérez-Pardo OC, Tovar-Doncel MS, Wissa S, Cimadevilla-Calvo B, Tejón-Pérez G an IT company: Medical Software Madrija Company which provided financial support in the form of a salary to Moreno-Fernández I and Escario-Méndez A; a statistician: Arévalo-Serrano J; an academic who help with the design: Valentín A and an external consultant who does not received any salary or fees: Lorenzo-Fernández JM. There are not currently competing financial interests or institutional conflicts among the authors of the study. The study is not linked to any company and there is not currently any competing interest associated such products in development, or marketed products. This does not alter our adherence to PLOS ONE policies on sharing data and materials. The rest of faculty and staff in a position to control or affect the content of this paper have declared that they have no competing financial interests or institutional conflicts. The rest of authors have declared that no competing interests exist.

## Results

The concordance rates between the interpretations of the clinicians and the ABG-a for acid-base disorders were an observed global agreement of 84,3% with a Cohen's kappa coefficient 0.81; 95% CI 0.77 to 0.86; p < 0.001. For detecting accuracy normal acid-base status the algorithm has a sensitivity of 90.0% (95% CI 79.9 to 95.3), a specificity 97.2% (95% CI 94.5 to 98.6) and a global accuracy of 95.9% (95% CI 93.3 to 97.6). For the four simple acid-base disorders, respiratory alkalosis: sensitivity of 91.2 (77.0 to 97.0), a specificity 100.0 (98.8 to 100.0) and global accuracy of 99.1 (97.5 to 99.7); respiratory acidosis: sensitivity of 61.1 (38.6 to 79.7), a specificity of 100.0 (98.8 to 100.0) and global accuracy of 98.0 (95.9 to 99.0); metabolic acidosis: sensitivity of 75.8 (59.0 to 87.2), a specificity of 99.7 (98.2 to 99.9) and a global accuracy of 97.4 (95.1 to 98.6); metabolic alkalosis sensitivity of 72.2 (56.0 to 84.2), a specificity of 95.5 (92.5 to 97.3) and a global accuracy of 93.0 (88.8 to 95.3); the four complex acid-base disorders, respiratory and metabolic alkalosis, respiratory and metabolic acidosis, respiratory alkalosis and metabolic acidosis, respiratory acidosis and metabolic alkalosis, the sensitivity, specificity and global accuracy was also high. For normal acid-base status the algorithm has PPV 87.1 (95% CI 76.6 to 93.3) %, and NPV 97.9 (95% CI 95.4 to 99.0) for a prevalence of 17.4 (95% CI 13.8 to 21.8). For the four-simple acid-base disorders and the four complex acid-base disorders the PPV and NPV were also statistically significant.

## Conclusions

The ABG-a showed very high agreement and diagnostic accuracy with experienced senior clinicians in the acid-base disorders in a clinical context. The method also provides refinement and deep complex analysis at the point-of-care that a clinician could have at the bedside on a day-to-day basis. The ABG-a method could also have the potential to reduce human errors by checking for imminent life-threatening situations, analysing the internal consistency of the results, the oxygenation and renal status of the patient.

## Introduction

The most imperative aspect of patients in emergency and critical care settings is their dynamic physiological status and potential for rapid deterioration that may require early diagnosis and clinical decisions for better patient outcome including pre-hospital medicine, using portable systems for the correct diagnosis of the patient's condition. Along with various "vital signs" such as, blood pressure, heart rate and rhythm, temperature, and respiratory rate, some biochemical markers reflect these rapid changes resulting in patient's unstable physiology [1]. Rapid provision of blood measurements, particularly blood gases and electrolytes, may translate into improved clinical outcomes. These situations require prompt lab results, most of which are done serially, ideally a point-of-care test (POCT), to meet the urgency of clinical decision and avoid subsequent damage to vital organs and systems [2]. Studies show that POCT carries advantages of providing reduced therapeutic turnaround time (TTAT), shorter door-to-clinical-decision time, rapid data availability, reduced pre-analytic and post-analytic testing errors, self-contained user-friendly instruments, small sample volume requirements, and frequent serial whole-blood testing [3,4]. This information provides vital in acute settings

**Fig 1. Standard arterial blood gas results from a commercially available analyser.** All current analysers, with more or less the same parameters, provide an output in the form of an on-screen or printed analysis which contains raw data, possibly with some standard reference values and indications as to whether the measured values fall inside or outside those reference values.

and patients on any type of extracorporeal support including extracorporeal membrane oxygenation (ECMO) especially on the initial set up until the patient is physiologically stable. This POCT is relevant also in chronic patients who attend on a regular basis to dialysis units or to respiratory outpatient clinics. In addition, the implementation and increased use of telemedicine in most countries is significant and thus the need for strong reliable software to support this new technology (e.g. supporting patients on home dialysis, clinics, small district/satellite hospitals, out-of-hours consultations, remote medicine, and district nurses) is essential.

A number of arterial blood gas (ABG) analysers from different manufactures exist which are commercially available. All such analysers, with more or less the same parameters, provide an output in the form of an on-screen or printed analysis which contains raw data, possibly with some standard reference values and indications as to whether the measured values fall inside or outside those reference values (Fig 1).

However, there is significant potential for human error when interpreting the blood measurements, which can be very complex on occasions. This, together with other stressors facing healthcare professionals today may affect the best possible clinical treatment of patients in a critical condition. Such stressors include: inexperienced non-laboratorians, reduced in hospital presence particularly out of hours of senior clinicians, ever increasing workloads of clinical staff, extreme working conditions as a result of the Covid-19 pandemic, financial burdens of tests, operator-dependent processes, reduced medical workforce, pre-hospital medicine and difficulty in integrating test results with hospital information systems (HIS) [3].

In order to reduce the impact of these stressors and improve patient safety the ABG-a is a recent technique designed for the automatic real-time analysis and interpretation of the arterial blood gases in a form of a written report, using only data from the arterial blood gases or when available results from the laboratory biochemistry department. ABG-a can also be connected to an interface to help with the communication and integration of the results with HIS and other devices (Figs 2 and 3).

The method was evaluated initially by the developers in Emergency Departments, Theatres and Critical Care settings in different hospitals with subpopulations of patients across all ages.

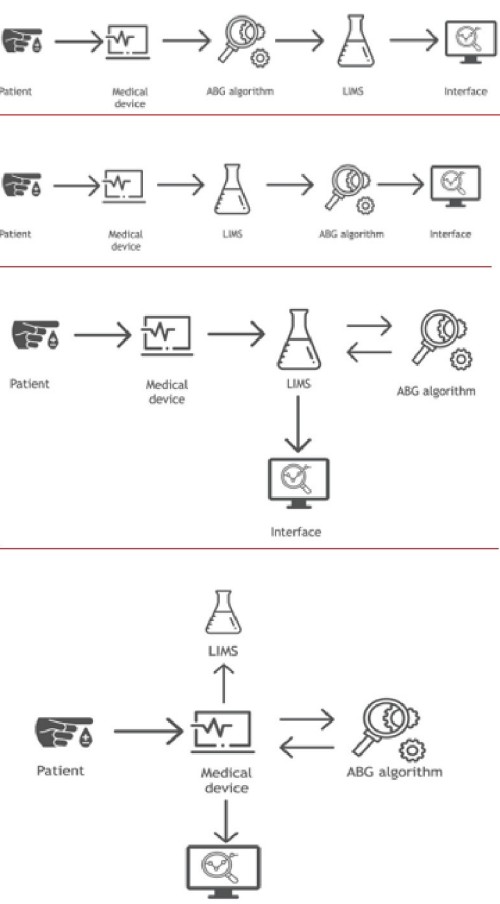

**Fig 2. Connectivity.** Data can be fed into the ABG-a manually for example, into small devices or used in countries where the digital hospital data has not yet been fully developed. It can also be embedded directly into medical devices, such as an arterial blood gas machine (including portable blood gas analysers), extracorporeal support machines, a monitor, database or any other biochemistry/haematology programs available through an interface for the communication and integration of the results with HIS and other devices.

The main aim of the present study is to evaluate the agreement and clinical usefulness of the ABG-a comparing the final software conclusion with the independent and unbiased diagnostic decision of senior clinicians among hospitalized patients, eligible for arterial blood assessment with acid-base disturbances interpretation. The second aim is to explore if the ABG-a could be considered a safety tool integrated into regular clinical practice by providing default calculations with the results.

## Materials and methods

### Study design and eligibility

This is a prospective multicentre, international, observational, cross sectional validation study: Complejo Hospitalario de Toledo (Spain), Marqués de Valdecilla University Hospital in Santander (Spain) and Medical University of Graz (Austria). The study conformed to the Declaration of Helsinki (a set of ethical principles regarding human experimentation developed for the medical community by the World Medical Association) and to local applicable regulatory requirements. The study was approved by each of the local Regional Research Ethics Committees [Comité de Ética de la Investigación con Medicamentos (CEIm) de Toledo, Spain; Comité

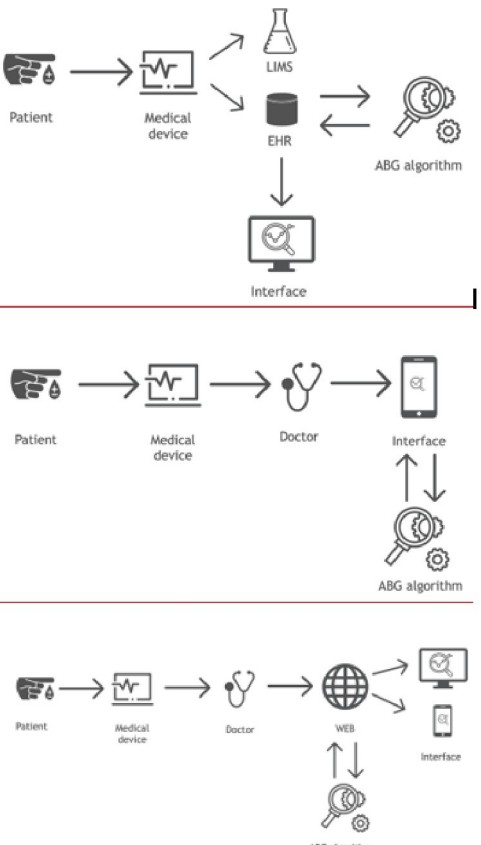

**Fig 3. Connectivity.** The communication and integration of the results with the hospital information system and other devices, can also be through the cloud and interface.

de Ética de la Investigación con Medicamentos (CEIm) de Cantabria, Spain and Vice-Rector for Research and International Affairs, Mag. a Caroline Schober-Trummler on behalf of the Med Uni Graz. (References 461, 2020.008 and 32-104ex19/20 respectively)]. All participants (or the next of kin/legally authorized representative in minors) in accordance with applicable law.

The article is reported in accordance with recommendations for non-randomized trials [5] and comparing diagnostic tests [6].

For accuracy, reproducibility and consistency, we standardized the inclusion and exclusion criteria used for patient recruitment. Subjects aged over two days old (full-term pregnancy) that for clinical reasons required invasive arterial monitoring during their stay in the Emergency Department, Theatres, Recovery or Critical Care Units between January 2020 and September 2020 were included in the study. The exclusion criteria included patients without inserted arterial lines or lines that were not functioning properly, samples with time between taking and analysing of more than 5 minutes; and evidence of errors in sampling, processing and analysis such as visible gas bubbles and/or blood clots. All participants (or the next of kin) provided written informed consent.

## Procedures and assessments

For the main study, patients admitted through the Emergency Department, Theatres, Recovery, interhospital transfers or any of the Critical Care Units were screened for eligibility by the responsible physician.

**Table 1. Classification of acid-base disturbances.**

| |
|---|
| 0 'Normal acid-base status' |
| 1 'Respiratory alkalosis' |
| 2 'Respiratory acidosis' |
| 3 'Metabolic acidosis' |
| 4 'Metabolic alkalosis' |
| 5 'Respiratory alkalosis and metabolic alkalosis' |
| 6 'Respiratory acidosis and metabolic acidosis' |
| 7 'Respiratory alkalosis and metabolic acidosis' |
| 8 'Respiratory acidosis and metabolic alkalosis' |
| 9 'I am not sure' |

Normal arterial blood gas values: pH 7,35–7,45; pCO2 4,7–6,0 KPa or pCO2 35-45mmHg; HCO3 22–30 (mean 26) mmol/L (= 26 mEq/L); SBE-2 to +2 mmol/L (-2 to +2 mEq/L); lactate 0,5–2,0 mmol/L (0,5–2,0 mEq/L). To convert from KPa to mmHg multiply by 7,5 (mmHg = KPa x 7,5).

ABG samples were taken by qualified nurses directly from the arterial catheter (ensuring the monitor displayed arterial waveforms properly) according to standard criteria. They were analysed straight away using the closest certified routine diagnostic methods on the instruments ABL800 FLEX (Radiometer Medical ApS, Brønshøj, Denmark) in Marqués de Valdecilla University Hospital; ABL-90 FLEX analyzer (Radiometer Medical ApS, Brønshøj, Denmark) in Medical University of Graz, and the GEM Premier 3000 system and GEM Premier 5000 system (Instrumentation Laboratory) in Hospital Virgen de la Salud.

Data including IDs of the patient and temperature corrected arterial blood gas samples and the date and times of each sample was recorded on a standardized study sheet. Acid-base disorders have been grouped into a total of nine categories (Table 1) from a total of 36 subcategories that the ABG-a can reproduce (Table in S1 Table).

In order to reduce internal variability three senior Consultant level clinicians, with more than ten years of postgraduate experience, were assigned in each centre for the assessment. Initially, each ABG was individually interpreted, initially by two different senior clinicians (A and B). A research ABG-a demo was downloaded as an application on the principal investigator device from each centre, for the software to interpret the results. The clinicians in charge of the patient did not have access to the software results and they were not able to use it in their clinical practice or decisions. Concordance was reached if the interpretations by the ABG-a coincided with the two clinicians (A and B); partial concordance was reached if the interpretations by the ABG-a coincided with one of the two clinicians' interpretations (A or B), in this case the two clinicians' interpretations and the interpretation by the ABG-a were compared separately against a third independent as external opinion (C). Final discordance was reached if the interpretations of the ABG-a did not agree with at least two of the three clinicians (A, B, C) (Table 2).

**Table 2.** *Questionnaire $C_I$* **= when software agrees with one clinician (A or B) but disagree with other (A or B).**

| Clinician A | Clinician B | Software | Clinician indept. ($C_I$) | S congruence |
|---|---|---|---|---|
| 1 | 1 | 1 | no | Yes |
| 1 | 1 | 2 | no | No |
| 1 | 2 | 1 | yes 1 | Yes |
| 1 | 2 | 1 | yes 2 | no |

Then a third and independent clinician ($C_I$) answered the questionnaire. *Questionnaire $A_R$ and $B_R$* = When software disagrees with any of the two clinicians or with both. The outcome of the software will not be used to suggest any treatment or any diagnosis in this study because it will not be available for the clinician. It is a blind study to the clinicians. The main objective is to compare the clinician's interpretation with the software interpretation.

### ABG-a

The ABG-a software by principle is an integrated mathematical algorithm, with foundations based on the current medical evidence for the analysis of oxygenation, acid-base and renal profiles. Some of the basis of this medical evidence have been previously published extensively elsewhere [7–11]. ABG-a has exclusive safety features such as an analysis of the internal consistency of the results with an identification of imminent life-threatening situations. Its preliminary diagnosis includes analysis of oxygenation, acid-base disturbances, blood urea nitrogen BUN (blood urea nitrogen)/creatinine ratio, and the URR (urea reduction ratio) as optional. The URR has been considered as useful in patients with presence of renal injury or renal replacement therapy [12]. Finally, ABG-a provides the clinician a differential physiological diagnosis and a potential list of causes of the identified disorder/s.

Data can be fed into the ABG-a manually for example, into small devices or used in countries where the digital hospital data has not yet been fully developed. It can also be embedded directly into medical devices, such as an arterial blood gas machine (including portable blood gas analysers), extracorporeal support machines, a monitor, database or any other biochemistry/haematology programs available through an interface for communication (Figs 2 and 3). The ABG-a software has been developed on a three-layer architecture to ensure that it can run under as many platforms as possible. The bottom layer of the software is the core algorithm, which is implemented in C++ and exchanges information in XML-format (Extensible Markup Language) with any other layer or piece of software.

The core algorithm can run on almost any device on the market, no matter whether it is Desktop, Mobile, Unix, Windows, cloud-based, on -premise or, even an embedded device. Furthermore, two different bridges, one for iOS devices to provide a software for testing, and a JNI (Java Native Interface) bridge has been developed, so it can interact with any piece of software written in Java. The described algorithm can also interact in seven different scenarios, with and without direct EHR/PDMS (Electronic Health Record)/(Patient Data Management System) integrations (Figs 2 and 3).

### Statistical analysis

Characteristics of patients and samples were tabulated as medians with range or interquartile range (IQR) for continuous variables. Categorical variables were expressed as frequencies and percentages.

The reliability also called agreement, reproducibility or consistency is the degree of coincidence of two or more measurements made on the same sample by one or more observers. The statistical tests used is the unweighted Cohen Kappa index with 95% asymptotic confidence interval for nominal measurements (also used was the observed agreement).

The diagnostic accuracy was evaluated with the index of sensitivity, specificity and efficiency of global accuracy. The clinical performance of the ABG-a was evaluated with the positive and negative predictive values for the prevalence of study population. The 95% confidence intervals of this diagnostic accuracy index were computed by de Wilson method. The analyses were performed using IBM SPSS Statistics 25.00 (IBM Corp., Armonk, NY, USA) and the Macro 'kappa for SPSS Statistics [13].

## Results

A preliminary version of ABG-a was tested initially with a total of 2348 calculations. A final ABG-a version of ABG-a was used for this validation study.

Between January and September 2020, 346 complete sample sets of consecutive ABG and biochemistry data (when available) was collected. The final analysis included 346 sample sets

from 64 patients. Median age was 63, IQR (interquartile range) 54 to 73 years; 38 (59.4%) were men, and the primary source of admission was Theatre or Recovery by any speciality. Total number of patients with one or more organs support (intubated, dialysis, etc.) were 30 (46.9%).

Clinical characteristics were retrieved by the principal investigator in each centre and they are shown in Table 3 (source admission), Table 4 (admitting diagnosis group) and Table 5 (chronic diseases and comorbidities), organ support and circumstances at the time the arterial sample was taken on the day of each test (Table 6).

## Clinical characteristics of patients

**Arterial blood gas and biochemistry profile values.** The temperature corrected arterial blood gas (or staple ticket) and the most recent biochemistry values (or staple results) of the 346 samples is shown in Tables 7 and 8 respectively.

## Reliability

The observed agreement and Cohen's kappa index between the interpretations of the experienced clinicians A, B and clinician results (A, B and C) with the ABG-a for acid-base disorders is shown in Table 9. For clinician results (A, B and C), observed agreement was 83.4% and Cohen's kappa index 0.81; 95% CI (0.77 to 0.86); $p < 0.001$ (Table 9).

## Diagnostic accuracy

For detecting with accuracy normal acid-base status the ABG-a has a sensitivity of 90.0% (95% CI 79.9 to 95.3), a specificity of 97.2% (95% CI 94.5 to 98.6) and a global accuracy of 95.9% (95% CI 93.3 to 97.6). For the four simple acid-base disorders (Respiratory alkalosis, Respiratory acidosis, Metabolic acidosis and Metabolic alkalosis) and the four complex acid- base

**Table 3. Type patients and source admission.**

|  | N | % |
|---|---|---|
| Age, years | 63[*] | 54 to 73[&] |
| Adult ($>$ = 16 years, there is no limit to high age) | 62 | 96.9 |
| Pediatric ($>$2 days to 16 years) | 2 | 3.1 |
| **Gender** |  |  |
| Male | 38 | 59.4 |
| Female | 26 | 40.6 |
| **Center** |  |  |
| Toledo | 29 | 45.3 |
| Santander | 20 | 31.3 |
| Austria | 15 | 23.4 |
| **Type of Admission** |  |  |
| Planned /elective post-theatre/Other planned | 38 | 59.4 |
| Unplanned/emergency | 26 | 40.6 |
| **Source of admission** |  |  |
| Theatre/recovery (any speciality) | 59 | 92.2 |
| Other hospital (transfer) | 3 | 4.7 |
| A&E | 2 | 3.1 |

[*]Mediam

[&]IQR (Interquartil range).

**Table 4. Admitting diagnosis group.**

|  | N | % |
|---|---|---|
| Cardiac arrest/other causes of cardiogenic shock | 1 | 1.6 |
| Cardiothoracic surgery | 9 | 14.1 |
| **Septic shock** | 29 | 45.3 |
| Surgical | 21 | 32.8 |
| Neutropenic sepsis/other hematology & oncology sepsis | 2 | 3.1 |
| Medical | 1 | 1.6 |
| Other | 5 | 7.8 |
| Post-operative management: any specialty (recovery anaesthesia) | 53 | 82.8 |
| **Haemorrhagic shock** | 8 | 12.5 |
| Gastrointestinal/other medical | 4 | 6.3 |
| Post-surgical | 4 | 6.3 |
| **Acute respiratory failure** | 7 | 10.9 |
| Pneumonia | 3 | 4.7 |
| Other causes* | 4 | 6.3 |
| Acute kidney injury or acute on chronic kidney disease | 7 | 10.9 |
| Trauma/neuro-surgical | 17 | 26.6 |
| Decompensated liver disease/acute liver failure | 3 | 4.7 |
| Diabetic ketoacidosis/hyperosmotic síndrome | 0 | 0.0 |
| Acute Intoxication of any type including medication, illegal drugs or herbs | 0 | 0.0 |
| Ulcerative colitis | 1 | 1.6 |

* No COPD /asthma exacerbation.

**Table 5. Chronic diseases and comorbidities.**

|  | N | % |
|---|---|---|
| Ischaemic heart disease | 6 | 9.4 |
| Congestive heart failure | 4 | 6.3 |
| Atrial fibrillation/Atrial flutter | 11 | 17.2 |
| Arterial hypertension | 8 | 12.5 |
| Pacemaker (PM) +/- Implantable cardioverter-defibrillator (ICD) | 3 | 4.7 |
| Cardiac resynchronization therapy (CRT) +/- (ICD) | 0 | 0.0 |
| Pulmonary hypertension | 0 | 0.0 |
| Pulmonary embolus/deep vein thrombosis | 0 | 0.0 |
| COPD/Asthma | 12 | 18.8 |
| Oxygen and/or non-invasive ventilation at home on regular bases | 3 | 4.7 |
| Pulmonary Fibrosis | 1 | 1.6 |
| Diabetes mellitus | 8 | 12.5 |
| Chronic kidney disease or pre-dialysis (Not on dialysis) | 4 | 6.3 |
| End-stage kidney disease (on regular dialysis) | 4 | 6.3 |
| Morbid obesity | 4 | 6.3 |
| Recent quimiotherapy /radiotherapy (last 30 days) | 2 | 3.1 |
| Acute myeloid leukemia (AML) | 1 | 1.6 |
| Other hemato-onco malignances | 0 | 0.0 |
| Use of diuretics and/or bicarbonate on regular bases | 7 | 10.9 |
| Any type of organ support | 30 | 46.9 |

**Table 6. Organ support/circumstances at the time the ABG was taken.**

| | N | % |
|---|---|---|
| Intravascular volume of the patient depleted | 49 | 14.2 |
| The $pO_2$ or the saturation increase with 100% $O_2$ | 346 | 100.0 |
| Hemofiltration, haemodialysis, MARS, PROMETHEUS, other | 1 | 0.3 |
| Intubated and on mechanical ventilation | 159 | 46.0 |
| Spontaneous ventilation on non-invasive ventilation | 7 | 2.0 |
| Spontaneous ventilation on high flow nasal cannula (Opti-flow) | 74 | 21.4 |
| Spontaneous (with oxygen) | 164 | 47.4 |
| Spontaneous (without oxygen) | 18 | 5.2 |
| ECMO, LVECD, full extracorporeal, other cardiac support devices | 3 | 0.9 |
| $CO_2$ removal device | 0 | 0.0 |
| Bicarbonate infusion or boluses given | 31 | 9.0 |
| Cardiovascular monitoring such as PiCCO, PAC (Pulmonar Artery Catheter, other.) | 27 | 7.8 |
| Other mechanical devices such as IABP, other. | 0 | 0.0 |
| During a cardiac arrest resuscitation | 0 | 0.0 |

disorders (Respiratory and metabolic alkalosis, Respiratory and metabolic acidosis, Respiratory alkalosis and metabolic acidosis, Respiratory acidosis and metabolic alkalosis) the sensitivity, specificity and global accuracy were also high (Table 10). The table also includes a column with the number of cases for each acid-base disorder in order to help with the interpretation. Since there are less positive cases than negative the sensitivity values are less than the specificity and the 95% CI are wider in the sensitivity and narrower in the specificity.

## Likelihood ratios

Likelihood ratios of the ABG-a analysis and interpretation of the ABG was shown in Table 12. For normal acid-base status the Positive Likelihood Ratio is 32.0, the Inverse Negative Likelihood Ratio is 9.7 (Negative Likelihood Ratio 0.1). Al the Positive and Inverse Negative Likelihood Ratios for the 4 simple and the 4 complex acid-base disorders are high and very high (Table 11).

## Clinical performance

For normal acid-base status the algorithm has Positive Predictive Value 87.1 (95% CI 76.6 to 93.3) %, and Negative Predictive Value 97.9 (95% CI 95.4 to 99.0) % for a prevalence of 17.4 (95% CI 13.8 to 21.8) %. For the four simple acid-base disorders (Respiratory alkalosis,

**Table 7. Corrected arterial blood gas (or staple ticket).**

| | Median | IQR |
|---|---|---|
| FiO2% | 40 | 28 to 40 |
| pH | 7.41 | 7.34 to 7.46 |
| pCO2 mmHg* | 39 | 35 to 43 |
| PO2 mmHg* | 106 | 83 to 138 |
| HCO3- mmol/L | 24.9 | 21.2 to 27.9 |
| SBE mmol/L | 0.05 | -3.70 to 3.60 |
| Lactate mmol/L | 1.16 | 0.78 to 2.00 |

*Coversion to kPa (mmHg/7.5).

**Table 8. Most recent biochemistry values (or staple results).**

|  | Median | IQR |
|---|---|---|
| Na mmol/L | 139 | 136 to 142 |
| K mmol/L | 4.1 | 3.7 to 4.5 |
| Cl mmol/L | 107 | 103 to 109 |
| Glucose mg/dL | 142 | 120 to 174 |
| Ca ionized mmol/L | 1.15 | 1.11 to 1.20 |
| Albumin g/dL | 2.2 | 3.1 to 3.6 |
| Urea* mg/dL | 49 | 36 to 77 |
| Creatinine mg/dL | 0.82 | 0.62 to 1.26 |

*Pre if dialysis or hemofiltration.

Respiratory acidosis, Metabolic acidosis and Metabolic alkalosis) and the four complex acid-base disorders (Respiratory and metabolic alkalosis, Respiratory and metabolic acidosis, respiratory alkalosis and metabolic acidosis, Respiratory acidosis and metabolic alkalosis the Positive and Negative Predictive Values were also high (Table 12).

## Discussion

As reported by the National Committee for Clinical Laboratory Standards, ABG analysis has a prospective influence on patient care over any other laboratory determinants [14]. The sudden changes in these parameters may result in life-threatening situations hence; rapid results are frequently required for effective management. POCT is of enormous help in pre-hospital emergency settings and have been employed for many years with immense success [15]. The management strategies in these life-threatening conditions immensely rely on rapid blood gas analysis [16]. Early detection of rapid clinical deterioration and associated changes in treatment is recommended by the World Health Organisation (WHO) and Surviving Sepsis Campaign guidelines [17,18]. POCT devices with rapid blood gas analysis are essentially required in these situations. Accurate and timely interpretation of an ABG can be lifesaving but establishing a correct interpretation and therefore to conclude with a diagnosis may be challenging as well as time consuming, with a risk of error in the calculations without an automated process.

Currently, most clinical laboratories and ABG machine manufacturers report only numeric values for the ABG results (Fig 1), and clinicians who are not specialists have difficulty interpreting the results and appropriately assessing a patient's status for pertinent therapeutic action.

Thus, interpretation of the ABG results depends on the judgement of experienced clinicians for a precise interpretation. However, whilst clinicians who request and receive raw data from patients' blood samples have considerable experience in this interpretation, they will not always be checking every aspect of the results and/or may overlook a particular aspect of the results if it was not part of the original reason for requesting ABG analysis. A thorough analysis

**Table 9. Fiability of the ABG-a for the analysis and interpretation of acid-base.**

|  | N | Observed agreement (%) | Kappa (95% CI) | p value |
|---|---|---|---|---|
| Clinician A | 344 | 81.7 | 0.79 (0.74 to 0.84) | < 0.001 |
| Clinician B | 342 | 74.6 | 0.71 (0.66 to 0.76) | < 0.001 |
| Clinicians Result (A, B and C) | 344 | 83.4 | **0.81 (0.77 to 0.86)** | **< 0.001** |

**Table 10. Diagnostic accuracy of automatic real-time analysis and interpretation of arterial blood sample.**

|  | N (%) | Sensitivity (95% CI) % | Specificity (95% CI) % | Global accuracy (95% CI) % |
|---|---|---|---|---|
| Normal acid-base status | 60 (17,4) | 90.0 (79.9 to 95.3) | 97.2 (94.5 to 98.6) | 95.9 (93.3 to 97.6) |
| Respiratory alkalosis | 34 (9,9) | 91.2 (77.0 to 97.0) | 100.0 (98.8 to 100.0) | 99.1 (97.5 to 99.7) |
| Respiratory acidosis | 18 (5,2) | 61.1 (38.6 to 79.7) | 100.0 (98.8 to 100.0) | 98.0 (95.9 to 99.0) |
| Metabolic acidosis | 33 (9,6) | 75.8 (59.0 to 87.2) | 99.7 (98.2 to 99.9) | 97.4 (95.1 to 98.6) |
| Metabolic alkalosis | 36 (10,5) | 72.2 (56.0 to 84.2) | 95.5 (92.5 to 97.3) | 93.0 (88.8 to 95.3) |
| Respiratory and metabolic alkalosis | 48 (14,0) | 79.2 (65.7 to 88.3) | 95.6 (92.6 to 97.4) | 93.3 (90.2 to 95.5) |
| Respiratory and metabolic acidosis | 51 (14,8) | 90.2 (79.0 to 95.7) | 97.3 (94.7 to 98.6) | 96.2 (93.6 to 97.8) |
| Respiratory alkalosis and metabolic acidosis | 31 (9,0) | 96.8 (83.8 to 99.4) | 98.1 (95.9 to 99.1) | 98.0 (95.8 to 99.0) |
| Respiratory acidosis and metabolic alkalosis | 31 (9,0) | 96.8 (83.8 to 99.4) | 98.1 (95.9 to 99.1) | 98.0 (95.9 to 99.0) |

can also be time consuming with a risk of error in the calculations without an automated process.

The ABG-a was designed to allow a detailed interpretation by the introduction of categories that reflect the extent of complexity for the four existing categories up to 36 categories. Given this, an algorithm for the automatic interpretation of ABG results would be useful for managing patients because it allows for the prompt and accurate interpretation of test results. We have assessed the validity of the algorithm by applying it to the interpretation of test results from clinical specimens. The main findings of our study are that ABG-a has very high agreement with the judgment of the senior clinician in all patients for the interpretation of acid-base status. This is the first published evaluation of ABG-a performed in independent centres from the developers. Strengths of the present study are that it includes a good sample size and a relevant target population on inpatients (emergency and elective) with a range of cardiopulmonary conditions and other pathologies eligible for blood gas measurement. Sampling was standardized and conducted by a dedicated specially trained nurse. The main analysis included only samples that met strict quality criteria. There were no signs of selection bias due to eligibility criteria as findings were robust when analysing all available samples. The analysis aligned to recent recommendations on the comparison of diagnostic tests [5,6].

At their broadest, aspects of the present method provide means of analysing an arterial blood sample, potentially at the point-of-care, which includes the following stages: safety analysis, oxygenation analysis, renal analysis and acid-base analysis. If any of the above sections are interpreted as abnormal or potential conditions identified, the software can then provide, for example, on screen or in printed form, a list of potential or likely causes and/or processes that explain the analytical findings.

**Table 11. Likelihood ratios of the ABG-a analysis and interpretation of acid-base.**

|  | Positive | Negative | Inverse Negative |
|---|---|---|---|
| Normal acid-base status | 32.0 | 0.1 | 9.7 |
| Respiratory alkalosis | * | 0.1 | 11.3 |
| Respiratory acidosis | * | 0.4 | 2.6 |
| Metabolic acidosis | 235.6 | 0.2 | 4.1 |
| Metabolic alkalosis | 15.9 | 0.3 | 3.4 |
| Respiratory and metabolic alkalosis | 18.0 | 0.2 | 4.6 |
| Respiratory and metabolic acidosis | 33.0 | 0.1 | 9.9 |
| Respiratory alkalosis and metabolic acidosis | 50.5 | 0.0 | 30.4 |
| Respiratory acidosis and metabolic alkalosis | 50.5 | 0.0 | 30.4 |

*No calculable, false positive is 0.0%.

**Table 12. Clinical performance of the ABG-a on the analysis and interpretation of acid-base.**

| | Positive Predictive Value (95% CI) % | Negative Predictive Value (95% CI) % | Prevalence (95% CI) % |
|---|---|---|---|
| Normal acid-base status | 87.1 (76.6 to 93.3) | 97.9 (95.4 to 99.0) | 17.4 (13.8 to 21.8) |
| Respiratory alkalosis | 100.0 (89.0 to 100.0) | 99.0 (97.2 to 99.7) | 9.9 (7.2 to 13.5) |
| Respiratory acidosis | 100.0 (74.1 to 100.0) | 97.9 (95.7 to 99.0) | 5.2 (3.3 to 8.1) |
| Metabolic acidosis | 96.2 (81.1 to 99.3) | 97.5 (95.1 to 98.7) | 9.6 (6.9 to 13.1) |
| Metabolic alkalosis | 65.0 (49.5 to 77.9) | 96.7 (94.1 to 98.2) | 10.5 (7.7 to 14.1) |
| Respiratory and metabolic alkalosis | 74.5 (61.1 to 84.5) | 96.6 (93.8 to 98.1) | 14.0 (10.7 to 18.0) |
| Respiratory and metabolic acidosis | 85.2 (73.4 to 92.3) | 98.3 (96.0 to 99.3) | 14.8 (11.5 to 19.0) |
| Respiratory alkalosis and metabolic acidosis | 83.3 (68.1 to 92.1) | 99.7 (98.2 to 99.9) | 9.0 (6.4 to 12.5) |
| Respiratory acidosis and metabolic alkalosis | 83.3 (68.1 to 92.1) | 99.7 (98.2 to 99.9) | 9.0 (6.4 to 12.5) |

## Safety

As an initial step, the ABG-a carries out a number of safety checks by default to alert the clinician to potential problems with the analysis. Included within these checks is an analysis of the internal consistency of the results. Errors in the measurement of plasma pH, $pCO_2$ or serum (total $CO_2/HCO_3^-$) are not uncommon [7]. If the values do not fit, this could suggest an error in one or more of the parameters and the measurements should be taken with caution. The alert includes advice to check on one or more of the following (non-exhaustive) sources of inconsistency such as: machine calibration; sample taken with a tourniquet; sample taken from a patient on dialysis with an A-V fistula with a tourniquet in place. The data collected from the study also showed clinicians do not routinely complete checks on numeric results for internal consistency. The alert may advise repeating the sample as a first check. A second safety check is for imminent life-threatening situations. This may include the checking of one or more of the $K^+$, $Na^+$, $Ca^{2+}$, lactate and/or glucose concentrations and/or the pH against thresholds [7].

## Oxygenation

If the patient is on mechanical ventilation, the ABG-a checks for hypoxemia by calculating the $PaO_2/FiO_2$ ratio on (PEEP 5cmH2O+) and if abnormal it will suggest or advise the clinician to rule out acute respiratory distress syndrome (ARDS) based on the Berlin definition [10]. If the patient is not on mechanical ventilation, the $pO_2$ is checked against normal values for group age and if abnormal, the A-a gradient is calculated and appropriate steps taken. From the questionnaire completed during the study, the Clinicians would welcome the routine automatic mathematical calculations of $PaO_2/FiO_2$ ratio and A-a gradient.

## Acid-base

The main limitation in order to be completely precise with the interpretation of the results from the ABG-a is the need of a clinical framework. The clinician should be aware that the assessment of an acid–base disorder is based on an accurate clinical evaluation and history. First, various signs and symptoms often provide clues regarding the underlying acid–base disorder; these include the patient's vital signs (which may indicate shock or sepsis), neurologic state (consciousness vs. unconsciousness), signs of infection (e.g., fever), pulmonary status (respiratory rate and presence or absence of Kussmaul respiration, cyanosis, and clubbing of the fingers), and gastrointestinal symptoms (vomiting and diarrhoea). Secondly, it is the time and conditions at which the sample was taken and thus subsequent pathological processed. E.g. the result at the onset of septic shock will differ from that at the end when the patient may be hyperventilating or receiving mechanical ventilator/dialysis support respectively. Thirdly,

certain underlying medical conditions such as liver, pregnancy, diabetes, and heart, lung, and kidney diseases may also affect the result. Finally, the clinician should determine whether the patient has taken any regular or new medications that affect acid–base balance (e.g. laxatives) and should consider signs of intoxication that may be associated with acid–base disturbances (e.g. acetone fetor as a sign of diabetic ketoacidosis).

In the physiological approach used by the ABG-a, a patient's acid–base status (meaning the physiologic derangement occurring at "a point in time") is classified according to one of the following four major acid-base disturbances defined as primary acid-base disorders: metabolic acidosis, metabolic alkalosis, respiratory acidosis and respiratory alkalosis. In many cases, however, a patient's acid–base status cannot be precisely classified into only one of the afore-mentioned four categories. Instead, test results may fall into a combination of two or more types of acid–base conditions and even one acid–base disturbance can present a broad range of test results, depending on the extent of the secondary response, compensatory respiratory or metabolic [7,19]. In many patients, we are unable to identify the order of presentation of clinical events or physiological derangement thus making it problematic to assign a label to the primary disorder and subsequent secondary response. These explanations may partially explain the relatively low sensitivities in respiratory acidosis (61.1%), metabolic acidosis (75.8%), and metabolic alkalosis (72.2%) results in our study (Table 10), all of which were iden-tified by the clinician as a "simple" acid-base disorder. Furthermore, in our opinion, the soft-ware begins to calculate as soon as values are identified outside the normal range, and therefore, obtaining more refined conclusions than that of the clinicians. The clinicians how-ever only considered the presence of "complex" acid-base disturbances when the results were obviously clearly deranged. This is likely due to multiple reasons including time pressures meaning not all ABG results can be rigorously examined with this approach.

Deviations from the prevailing normal value of the serum anion gap can reflect either errors in the measurements of its constituents or changes in the concentrations of unmeasured cat-ions and/or anions. Given this wide inter-individual variability, it is important, where possible, to know the prevailing baseline value of the serum anion gap for a particular individual [20]. If the baseline serum anion gap of an individual is not known and the range of normal values of a particular laboratory is used to assess the anion gap, then it is possible that disorders that cause deviations in the serum anion gap might not be recognized since they are insufficient to shift the serum anion gap outside the normal range [20].

The advantage of the ABG-a is that it can successfully provide the identification and description of the acid-base disturbances and secondary responses associated from each sam-ple and has the additional benefit of taking into account the basal anion gap and bicarbonate values to overcome the problems explained. Nevertheless, the clinician must review all results taking into account the clinical context of the situation they are presented with.

The concept of the automatic interpretation of acid-base using a computer program is not new. Since the first publications in the 1980's [21–23], several web-based programs have been developed and are available online. However, the lack of a description of material and methods in most cases and a standard validation methodology make the comparison among studies and web-based programs impossible.

## Renal analysis

The ABG-a analyses the BCR (BUN Creatinine ratio) automatically when the electrolytes, from either from ABG or/and HIS, are available. If it is not within the normal range, depend-ing on what side of the range it lies, a selection of possible causes are identified. If the patient is on dialysis or CRRT (continuous renal replacement therapy), the algorithm can calculate the

urea reduction ratio (URR) and determine if the dialysis is grossly performing adequately. The Kt/V is mathematically related to the URR (on average, a Kt/V of 1.2 is roughly equivalent to a URR of about 63 percent). The ABG-a can provide an initial estimation of the renal support treatment at POC. It was not the aim of this study to validate these features since they are mathematical calculations. The Clinicians reported via the study's questionnaire that routine automatic calculation of BCR would be welcomed in order to obtain more initial information at Point-of-Care (POC).

The further benefit is the possibility of the ABG-a to be used outside of acute settings particularly for patients with chronic conditions such as long term respiratory pathologies e.g. chronic obstructed pulmonary disease (COPD) or renal patients on dialysis. This may provide reliable clinical interpretation with the option of taking into account known basal values of the anion gap and bicarbonate at the time of the analysis. Many of these patients are well known to the respiratory outpatients clinics or satellite dialysis units. The ABG-a could be a potential tool in the screening for an acute on chronic decompensation secondary to any other pathological process, to follow up clinical progression, monitoring during dialysis sessions or even telemedicine through application of the software to portable devices [21,24].

## Conclusions

The ABG-a showed very high agreement and diagnostic accuracy with experienced senior clinicians in the acid-base disorders in a clinical context. The method also provides refinement and deep complex analysis for the point-of-care that a clinician could have at the bedside on a day-to-day basis. The ABG-a method could also have the potential to reduce human errors by checking for imminent life-threatening situations, analysing the internal consistency of the results and the oxygenation and renal statuses of the patient.

## Supporting information

**S1 Table. ABG-a diagnosis with 36 categories.**
(DOCX)

**S1 Text. Three clinical cases to ilustrate the ABG-a use.**
(DOCX)

**S1 Raw data Raw blood gas data 06_02_21.**
(XLSX)

## Acknowledgments

The authors thank the patients and, in some cases, the next of kin who made this research possible.

## Author Contributions

**Conceptualization:** Sancho Rodríguez-Villar, Bruno Manuel Do-Vale, Helen Marie Fletcher.

**Data curation:** Sancho Rodríguez-Villar, Paloma Poza-Hernández, Sascha Freigang, Idoia Zubizarreta-Ormazabal, Daniel Paz-Martín, Etienne Holl, Osvaldo Ceferino Pérez-Pardo, María Sherezade Tovar-Doncel, Sonja Maria Wissa, Bonifacio Cimadevilla-Calvo, Guillermo Tejón-Pérez.

**Formal analysis:** Sancho Rodríguez-Villar, Juan Arévalo-Serrano.

**Funding acquisition:** Sancho Rodríguez-Villar.

**Investigation:** Sancho Rodríguez-Villar, Paloma Poza-Hernández, Sascha Freigang, Idoia Zubizarreta-Ormazabal, Daniel Paz-Martín, Etienne Holl, Osvaldo Ceferino Pérez-Pardo, María Sherezade Tovar-Doncel, Sonja Maria Wissa, Bonifacio Cimadevilla-Calvo, Guillermo Tejón-Pérez.

**Methodology:** Sancho Rodríguez-Villar, Juan Arévalo-Serrano, Antonio Valentín.

**Project administration:** Sancho Rodríguez-Villar, Paloma Poza-Hernández, Sascha Freigang, Idoia Zubizarreta-Ormazabal.

**Resources:** Sancho Rodríguez-Villar, Paloma Poza-Hernández, Sascha Freigang, Idoia Zubizarreta-Ormazabal.

**Software:** Sancho Rodríguez-Villar, Ismael Moreno-Fernández, Alejandro Escario-Méndez.

**Supervision:** Sancho Rodríguez-Villar, Juan Arévalo-Serrano, Antonio Valentín.

**Validation:** Juan Arévalo-Serrano, Antonio Valentín.

**Visualization:** Sancho Rodríguez-Villar.

**Writing – original draft:** Sancho Rodríguez-Villar, Bruno Manuel Do-Vale, Helen Marie Fletcher, Jesús Medardo Lorenzo- Fernández.

**Writing – review & editing:** Sancho Rodríguez-Villar, Paloma Poza-Hernández, Sascha Freigang, Idoia Zubizarreta-Ormazabal, Daniel Paz-Martín, Etienne Holl, Osvaldo Ceferino Pérez-Pardo, María Sherezade Tovar-Doncel, Sonja Maria Wissa, Bonifacio Cimadevilla-Calvo, Guillermo Tejón-Pérez, Ismael Moreno-Fernández, Alejandro Escario-Méndez, Juan Arévalo-Serrano, Antonio Valentín, Bruno Manuel Do-Vale, Helen Marie Fletcher, Jesús Medardo Lorenzo- Fernández.

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
