## [Decision Letter · Decision Letter 0]

4 Dec 2020

PONE-D-20-32648

Automatic real-time analysis and interpretation of arterial blood gas for Point-of-Care testing: clinical validation

PLOS ONE

Dear Dr. Rodriguez-Villar,

Thank you for submitting your manuscript to PLOS ONE. After careful consideration, we feel that it has merit but does not fully meet PLOS ONE’s publication criteria as it currently stands. Therefore, we invite you to submit a revised version of the manuscript that addresses the points raised during the review process.

We look forward to receiving your revised manuscript.

Kind regards,

Tai-Heng Chen, M.D.

Academic Editor

PLOS ONE

2. Thank you for including your ethics statement:  "This is a prospective multicentre, international, observational, cross sectional validation study: Complejo Hospitalario de Toledo (Spain), Marqués de Valdecilla University Hospital in Santander (Spain) and Medical University of Graz (Austria). The study conformed to the Declaration of Helsinki (a set of ethical principles regarding human experimentation developed for the medical community by the World Medical Association) and to local applicable regulatory requirements. The study was approved by each of the local Regional Research Ethics Committees (references 461, 2020.008 and 32-104ex19/20 respectively). All participants (or the next of kin/ legally authorized representative in minors) in accordance with applicable law.".   

2. Please ensure you have thoroughly discussed any potential limitations of this study within the Discussion section, including the potential impact of confounding factors.

"All faculty and staff in a position to control or affect the content of this paper have declared that they have no competing financial interests or institutional conflicts. The authors have declared that no competing interests exist."

We note that one or more of the authors are employed by a commercial company: Madrija Company, External Consultant. Ingeniero de Caminos.

3.1. Please provide an amended Funding Statement declaring this commercial affiliation, as well as a statement regarding the Role of Funders in your study. If the funding organization did not play a role in the study design, data collection and analysis, decision to publish, or preparation of the manuscript and only provided financial support in the form of authors' salaries and/or research materials, please review your statements relating to the author contributions, and ensure you have specifically and accurately indicated the role(s) that these authors had in your study. You can update author roles in the Author Contributions section of the online submission form.

3.2. Please also provide an updated Competing Interests Statement declaring this commercial affiliation along with any other relevant declarations relating to employment, consultancy, patents, products in development, or marketed products, etc.  

6. Please include a separate caption for each figure in your manuscript.

Reviewers' comments:

Reviewer's Responses to Questions

**Comments to the Author**

1. Is the manuscript technically sound, and do the data support the conclusions?

Reviewer #1: Yes

Reviewer #2: Partly

2. Has the statistical analysis been performed appropriately and rigorously? 

Reviewer #1: Yes

Reviewer #2: Yes

3. Have the authors made all data underlying the findings in their manuscript fully available?

Reviewer #1: Yes

Reviewer #2: No

4. Is the manuscript presented in an intelligible fashion and written in standard English?

Reviewer #1: Yes

Reviewer #2: No

5. Review Comments to the Author

Reviewer #1: Based on the results, the arterial blood gas algorithm (ABG-a) is practical for acute and critical care. Only one questions are needed for explanation and one suggestion are recommended.

Question: Please explain the relatively low sensitivity in respiratory acidosis (61.1%), metabolic acidosis (75.8%), and metabolic alkalosis (72.2%).

Suggestion: The data (median & IQR) from enrolled samples in Table 7 & 8 seem so normal. The reports of minimum and maximum levels of tests may be demonstrated, as a kind of description of the severity and variability of tested samples.

Some errors need to correct.

Table 4: Acute renal failure or acute on chronic renal failure � acute kidney injury or acute on chronic kidney disease

Table 8: 4,1  4.1

Line 352: 83,4% 83.4%

Reviewer #2: In this manuscript, authors compare the results of an automated acid-base disturbance algorithm to the opinions of experienced physicians. Acid base disorders are complicated, integrating many clinical observations, diagnosis, and recent treatment history into a mechanism underlying physiological perturbations. Thus the clinical relevance of an algorithmic solution is established.

As this manuscript is not describing the algorithm, but only its agreement with clinicians, there are some points that I think would add tremendous value to the work. Chief among them is a calculation of the interpretability of the algorithm. In machine learning, there are several tools that allow a user to test how a classification changes, among them ICE (individual continuous expectation) and partial dependence plots. These tools let you assess how a small change in one measurement would affect the algorithmic diagnosis. Other uncertainty quantification methods may be used instead, but some measure of interpretability may be useful.

Additionally, reporting the global accuracy in this type of work is misleading. The dataset is unbalanced (many more negative cases that positive cases, for at least some of the choices of acid base disorder. Without knowledge of the actual numbers of diagnoses (how many negative, how many metabolic acidosis, etc) it is impossible to interpret global accuracy, and its difficult to assess the importance of poor performance on some disease states. I would request that the authors include the number of patients with each disorder; this would be part of the PLOS requirement that "all data be made available". Additionally, reporting a beta-weighted F-score may give a more interpretable level of global accuracy, with beta chosen to reflect the relative cost between a false negative and a false positive.

There are some parts of the manuscript where English usage and idiom take away from the results being presented. Manuscript would be improved with editing.

6. PLOS authors have the option to publish the peer review history of their article (what does this mean?). If published, this will include your full peer review and any attached files.

Reviewer #1: No

Reviewer #2: No

---

## [Author Response · Author response to Decision Letter 0]

18 Dec 2020

12/12/20

REBUTTAL LETTER

Dear Academic Editor and Reviewers,

RE: PONE-D-20-32648

Automatic real-time analysis and interpretation of arterial blood gas for Point-of-Care testing: clinical validation

Thank you for reviewing and providing feedback to the paper. We will try to answer all your comments in a rigorous fashion by order.

Reviewer #1: Based on the results, the arterial blood gas algorithm (ABG-a) is practical for acute and critical care. Only one question are needed for explanation and one suggestion are recommended.

Question: Please explain the relatively low sensitivity in respiratory acidosis (61.1%), metabolic acidosis (75.8%), and metabolic alkalosis (72.2%).

From the eight acid-base disorders, only three of them have a sensitivity of less than 80%. In the diagnosis of acid-base disorders, there is really no diagnostic test considered as reference “Gold standard” except for an expert opinion.

Firstly, it is very remarkable that only this relatively low sensitivity occurs with those acid-base disorders classified as “single” and not “complex” acid-base disorders. In our opinion based on the observations during the study, we can appreciate that the software is able to run calculations on extremely mild deranged values from the arterial blood gases, and therefore can obtain more refined conclusions than that of the clinicians. Also, the ABG-a follows the rules of certain values cutting points. The clinicians however only consider the presence of “complex” acid-base disturbances when the results are very obviously deranged. This approach from the clinicians maybe explained by many reasons, not least of all limited time for analysing every single ABG result in such a rigorous and complex manner. The diagnostic judgment of experienced clinicians has been taken as a reference test, which without denying its validity, can change from one clinician to another. Further studies in this regard would be necessary, after which it would be possible to consider our algorithm as a reference test.

Secondly, those diagnosis made by the ABG-a as “complex” still include the primary diagnosis of “respiratory acidosis”, “metabolic acidosis” and “metabolic alkalosis”. Not only keeping the safety standards but also improving the diagnosis. We know that in the human physiology, the presence of disturbances are complex and frequently not simple, making the ABG-a of great value in order to help with a more refined diagnosis.

Suggestion: The data (median & IQR) from enrolled samples in Table 7 & 8 seem so normal. The reports of minimum and maximum levels of tests may be demonstrated, as a kind of description of the severity and variability of tested samples.

We could understand that the suggestion relates to utilising means and standard deviations as opposed to medians and interquartile ranges because this form still present in some publications, but asking for the maximum and minimum values to describe and represent a quantitative variable is in our opinion inappropriate. The current trend is to describe quantitative values with median and interquartile range because they do not require compliance with normality; they are very robust, represent the data well and are not affected by the presence of extreme values. On the contrary, if they were described with the range and the minimum and maximum values, the representativeness of the data would be violated since those minimum and maximum values are usually atypical and remote values.

Some errors need to correct:

Table 4: Acute renal failure or acute on chronic renal failure � acute kidney injury or acute on chronic kidney disease

Table 8: 4,1  4.1

Line 352: 83,4% 83.4%

We agree and these errors have been corrected on the manuscript.

Reviewer #2:

Additionally, reporting the global accuracy in this type of work is misleading. The dataset is unbalanced (many more negative cases than positive cases, for at least some of the choices of acid base disorder. Without knowledge of the actual numbers of diagnoses (how many negative, how many metabolic acidosis, etc) it is impossible to interpret global accuracy, and it is difficult to assess the importance of poor performance on some disease states. I would request that the authors include the number of patients with each disorder; this would be part of the PLOS requirement that "all data be made available". Additionally, reporting a beta-weighted F-score may give a more interpretable level of global accuracy, with beta chosen to reflect the relative cost between a false negative and a false positive.

We agree that reporting global accuracy can be misleading and that reporting the frequency of each of the acid-base disorders helps with interpretation of the data. Table 10 has been modified to include a column with the number of cases of each of the acid-base disorders. It is observed that since there are fewer positive than negative cases, the sensitivity values are lower than those of specificity and the 95% CIs are wider in sensitivity and narrower in specificity. By reporting the global accuracy together with the sensitivity and specificity as well as the frequency of each disorder, as reflected in the modified table 10, there is less misunderstanding. The beta-weighted F-score could have been used as suggested, although it has the drawbacks that it ignores true negatives in its calculation and assigns equal importance to precision and recall. In our work we have chosen the analysis of reliability with the kappa index and accuracy with sensitivity, specificity and, together with them, and not in isolation, the global accuracy.

PLOS authors have the option to publish the peer review history of their article (what does this mean?). If published, this will include your full peer review and any attached files.

We are agreed as authors to be published our article peer review history

Best regards

Dr Sancho Rodriguez-Villar

---

## [Decision Letter · Decision Letter 1]

5 Feb 2021

PONE-D-20-32648R1

Automatic real-time analysis and interpretation of arterial blood gas for Point-of-Care testing: clinical validation

PLOS ONE

Dear Dr. Rodriguez-Villar,

Thank you for submitting your manuscript to PLOS ONE. After careful consideration, we feel that it has merit but does not fully meet PLOS ONE’s publication criteria as it currently stands. Therefore, we invite you to submit a revised version of the manuscript that addresses the points raised during the review process.

We look forward to receiving your revised manuscript.

Kind regards,

Tai-Heng Chen, M.D.

Academic Editor

PLOS ONE

Reviewers' comments:

Reviewer's Responses to Questions

**Comments to the Author**

1. If the authors have adequately addressed your comments raised in a previous round of review and you feel that this manuscript is now acceptable for publication, you may indicate that here to bypass the “Comments to the Author” section, enter your conflict of interest statement in the “Confidential to Editor” section, and submit your "Accept" recommendation.

Reviewer #1: All comments have been addressed

Reviewer #3: All comments have been addressed

2. Is the manuscript technically sound, and do the data support the conclusions?

Reviewer #1: Partly

Reviewer #3: Yes

3. Has the statistical analysis been performed appropriately and rigorously? 

Reviewer #1: Yes

Reviewer #3: I Don't Know

4. Have the authors made all data underlying the findings in their manuscript fully available?

Reviewer #1: Yes

Reviewer #3: No

5. Is the manuscript presented in an intelligible fashion and written in standard English?

Reviewer #1: Yes

Reviewer #3: Yes

6. Review Comments to the Author

Reviewer #1: (No Response)

Reviewer #3: I think this paper is very interesting and well written. But, I have only one serious concern.

According to Tables 7 and 8, both median and IQR are in the normal range. What are the values of the raw data that is the basis of this analysis? The author should show raw blood gas data as shown in PLOS Data policy.

7. PLOS authors have the option to publish the peer review history of their article (what does this mean?). If published, this will include your full peer review and any attached files.

Reviewer #1: No

Reviewer #3: No

---

## [Author Response · Author response to Decision Letter 1]

9 Feb 2021

06/02/21

REBUTTAL LETTER

Dear Academic Editor and Reviewers,

RE: PONE-D-20-32648

Automatic real-time analysis and interpretation of arterial blood gas for Point-of-Care testing: clinical validation

Thank you for reviewing and providing feedback to the paper. We will try to answer all your comments in a rigorous fashion by order.

6. Review Comments to the Author

Reviewer #1: (No Response)

Reviewer #3: I think this paper is very interesting and well written. But, I have only one serious concern.

According to Tables 7 and 8, both median and IQR are in the normal range. What are the values of the raw data that is the basis of this analysis? The author should show raw blood gas data as shown in PLOS Data policy.

Anonymous raw blood gas data showed as per PLOS Data policy. Please, see attached .

Best regards

Dr Sancho Rodriguez-Villar

---

## [Decision Letter · Decision Letter 2]

17 Feb 2021

PONE-D-20-32648R2

Automatic real-time analysis and interpretation of arterial blood gas for Point-of-Care testing: clinical validation

PLOS ONE

Dear Dr. Rodriguez-Villar,

Thank you for submitting your manuscript to PLOS ONE. After careful consideration, we feel that it has merit but does not fully meet PLOS ONE’s publication criteria as it currently stands. Therefore, we invite you to submit a revised version of the manuscript that addresses the points raised during the review process.

We look forward to receiving your revised manuscript.

Kind regards,

Tai-Heng Chen, M.D.

Academic Editor

PLOS ONE

Reviewers' comments:

Reviewer's Responses to Questions

**Comments to the Author**

1. If the authors have adequately addressed your comments raised in a previous round of review and you feel that this manuscript is now acceptable for publication, you may indicate that here to bypass the “Comments to the Author” section, enter your conflict of interest statement in the “Confidential to Editor” section, and submit your "Accept" recommendation.

Reviewer #3: All comments have been addressed

2. Is the manuscript technically sound, and do the data support the conclusions?

Reviewer #3: Yes

3. Has the statistical analysis been performed appropriately and rigorously? 

Reviewer #3: Yes

4. Have the authors made all data underlying the findings in their manuscript fully available?

Reviewer #3: Yes

5. Is the manuscript presented in an intelligible fashion and written in standard English?

Reviewer #3: Yes

6. Review Comments to the Author

Reviewer #3: My concerns have been resolved. But, I don't know if the raw blood gas data does not have to be attached to the Manuscript. Leave it to the editor's judgment.

7. PLOS authors have the option to publish the peer review history of their article (what does this mean?). If published, this will include your full peer review and any attached files.

Reviewer #3: No

---

## [Author Response · Author response to Decision Letter 2]

21 Feb 2021

Dear Editor, 

We are delighted that the Reviewer found our work to be of interest and value. We are grateful to him/her for taking time to comment on our manuscript and address each of the queries on a point-by-point basis. We think that the amended manuscript is improved and do very much hope that it is now considered acceptable for publication. 

Our answers to the comment from Reviewer #3: follow below. 

Reviewer #3: My concerns have been resolved. But I don't know if the raw blood gas data does not have to be attached to the Manuscript. Leave it to the editor's judgment.

Answer: From us there is no problem to add the raw arterial blood gas data. As authors we give consent and we leave that decision to the Editor-in -Chief.

Many thanks for your time.

Dr. Sancho Rodríguez-Villar 

Honorary Senior Clinical Lecturer 

Department of Basic and Clinical Neuroscience.

Institute of Psychiatry, Psychology & Neuroscience (IoPPN)

Academic Neuroscience Centre, 

London SE5 8AF, UK

T: 0207 848 0293

---

## [Decision Letter · Decision Letter 3]

24 Feb 2021

Automatic real-time analysis and interpretation of arterial blood gas for Point-of-Care testing: clinical validation

PONE-D-20-32648R3

Dear Dr. Rodriguez-Villar,

We’re pleased to inform you that your manuscript has been judged scientifically suitable for publication and will be formally accepted for publication once it meets all outstanding technical requirements.

Kind regards,

Tai-Heng Chen, M.D.

Academic Editor

PLOS ONE

Reviewers' comments:

Reviewer's Responses to Questions

**Comments to the Author**

1. If the authors have adequately addressed your comments raised in a previous round of review and you feel that this manuscript is now acceptable for publication, you may indicate that here to bypass the “Comments to the Author” section, enter your conflict of interest statement in the “Confidential to Editor” section, and submit your "Accept" recommendation.

Reviewer #3: All comments have been addressed

2. Is the manuscript technically sound, and do the data support the conclusions?

Reviewer #3: Yes

3. Has the statistical analysis been performed appropriately and rigorously? 

Reviewer #3: Yes

4. Have the authors made all data underlying the findings in their manuscript fully available?

Reviewer #3: Yes

5. Is the manuscript presented in an intelligible fashion and written in standard English?

Reviewer #3: Yes

6. Review Comments to the Author

Reviewer #3: My concerns have been resolved. But I don't know if the raw blood gas data does not have to be attached to the Manuscript. Leave it to the editor's judgment.

7. PLOS authors have the option to publish the peer review history of their article (what does this mean?). If published, this will include your full peer review and any attached files.

Reviewer #3: No

---

## [Editor Report · Acceptance letter]

25 Feb 2021

PONE-D-20-32648R3 

Automatic real-time analysis and interpretation of arterial blood gas sample for Point-of-Care testing: clinical validation 

Dear Dr. Rodriguez-Villar:

I'm pleased to inform you that your manuscript has been deemed suitable for publication in PLOS ONE. Congratulations! Your manuscript is now with our production department. 

Kind regards, 

on behalf of

Dr. Tai-Heng Chen 

Academic Editor

PLOS ONE